# MicroRNA Expression in Clear Cell Renal Cell Carcinoma Cell Lines and Tumor Biopsies: Potential Therapeutic Targets

**DOI:** 10.3390/ijms23105604

**Published:** 2022-05-17

**Authors:** Samuel Swearson, Aseel O. Rataan, Steven Eliason, Brad A. Amendt, Yousef Zakharia, Aliasger K. Salem, Thai Ho, Youcef M. Rustum

**Affiliations:** 1Department of Anatomy and Cell Biology, Craniofacial Anomalies Research Center, The University of Iowa, Iowa City, IA 52242, USA; samuel-swearson@uiowa.edu (S.S.); steven-eliason@uiowa.edu (S.E.); brad-amendt@uiowa.edu (B.A.A.); 2Department of Pharmaceutical Sciences and Experimental Therapeutics, College of Pharmacy, University of Iowa, Iowa City, IA 52242, USA; aseel-rataan@uiowa.edu (A.O.R.); aliasger-salem@uiowa.edu (A.K.S.); 3Department of Internal Medicine, Holden Comprehensive Cancer Center, University of Iowa, Iowa City, IA 52242, USA; yousef-zakharia@uiowa.edu; 4Department of Internal Medicine, Hematology/Oncology, Mayo Clinic, Phoenix, AZ 85259, USA; ho.thai@mayo.edu; 5Department of Internal Medicine, Carver College of Medicine, University of Iowa, Iowa City, IA 52242, USA; 6Department of Pharmacology & Therapeutics, Roswell Park Comprehensive Cancer Center, Buffalo, NY 14203, USA

**Keywords:** microRNAs, clear cell renal cell carcinoma, sarcomatoid ccRCC, primary human kidney biopsies with matching normal kidney tissue

## Abstract

This study was carried out to quantitate the expression levels of microRNA-17, -19a, -34a, -155, and -210 (miRs) expressed in nine clear cell renal cell carcinoma (ccRCC) and one chromophobe renal cell carcinoma cell line with and without sarcomatoid differentiation, and in six primary kidney tumors with matching normal kidney tissues. The data in the five non-sarcomatoid ccRCC cell lines—RC2, CAKI-1, 786-0, RCC4, and RCC4/VHL—and in the four ccRCC with sarcomatoid differentiation—RCJ41T1, RCJ41T2, RCJ41M, and UOK-127—indicated that miR-17 and -19a were expressed at lower levels relative to miR-34a, -155, and -210. Compared with RPTEC normal epithelial cells, miR-34a, miR-155, and miR-210 were expressed at higher levels, independent of the sarcomatoid differentiation status and hypoxia-inducible factors 1α and 2α (HIFs) isoform expression. In the one chromophobe renal cell carcinoma cell line, namely, UOK-276 with sarcomatoid differentiation, and expressing tumor suppressor gene *TP53*, miR-34a, which is a tumor suppressor gene, was expressed at higher levels than miR-210, -155, -17, and -19a. The pilot results generated in six tumor biopsies with matching normal kidney tissues indicated that while the expression of miR-17 and -19a were similar to the normal tissue expression profile, miR-210, -155, -and 34a were expressed at a higher level. To confirm that differences in the expression levels of the five miRs in the six tumor biopsies were statistically significant, the acquisition of a larger sample size is required. Data previously generated in ccRCC cell lines demonstrating that miR-210, miR-155, and HIFs are druggable targets using a defined dose and schedule of selenium-containing molecules support the concept that simultaneous and concurrent downregulation of miR-210, miR-155, and HIFs, which regulate target genes associated with increased tumor angiogenesis and drug resistance, may offer the potential for the development of a novel mechanism-based strategy for the treatment of patients with advanced ccRCC.

## 1. Introduction

Although mechanisms of drug resistance and increased angiogenesis in molecularly and immunologically heterogeneous tumors are multifactorial, oncogenic miR-210 and -155 (miRs), in addition to HIF1α and HIF2α, are reportedly overexpressed in many advanced cancers, regulating multiple targets involved in tumor growth, mitochondrial function, and response to anticancer therapy [1,2,3,4,5,6,7,8]. miRs and HIFs are upregulated in cancer cells, regardless of normoxic or hypoxic conditions, and directly or indirectly regulate the expression of several target genes, including transforming growth factor-beta (TGF-β); the nuclear factor-erythroid factor 2-related factor 2 (Nrf2), which regulates the expression of programmed death-ligand 1 (PD-L1); and vascular endothelial growth factor-A (VEGF-A) in clear cell renal cell carcinoma (ccRCC) tumors with and without sarcomatoid differentiation [8,9,10,11]. TGF-β regulates, in part, the expression of Nrf2, PD-L1, VEGF, and immune response targets associated with resistance, including forkhead box P3 (Foxp3), regulatory T cells (Treg), and myeloid-derived suppressor cells (MDSC) [9,10,11,12,13,14,15,16,17,18,19]. Although sarcomatoid features are expressed in all renal cell carcinoma phenotypes, 5–15% of ccRCC tumors express sarcomatoid cells. These tumors respond poorly to standard therapies and express a unique molecular profile.

The non-coding miRs with an average of 18–24 nucleotides in length are expressed in many advanced cancers and play a critical role in the post-translational regulation of target genes implicated in tumor growth, metastasis, angiogenesis, and drug resistance [4,5,6,7,19,20,21,22,23,24,25,26]. Specific types of miRs are expressed in normal tissues as tumor suppressors, offering protection against oxidative stress. However, in advanced tumor tissues, the altered miRs may function as oncogenes. miR-210 and -155 are the most investigated miRs, are ubiquitously overexpressed in the majority of advanced solid tumors and could potentially serve as a critical therapeutic target. In many tumors, miRs and HIFs are co-expressed and regulate common and independent targets [27,28,29,30,31,32,33]. Inhibitors of HIF2α are under clinical development with promising efficacy [34,35]. The oncogenic miRs function via an interaction with the 3′ untranslated region of target messenger RNAs (mRNAs), resulting in mRNA degradation and/or the translational repression of target genes. Although considerable efforts are underway for the development of miR inhibitors and/or activators, the toxicity, efficacy, and delivery continue to represent significant challenges [36,37,38,39,40,41,42].

Our laboratory was the first to demonstrate that miR-210, miR-155, HIFs, PD-L1, VEGF, Nrf2, and more recently TGF-β, are selenium targets and can be downregulated using therapeutic doses of seleno-L-methionine (SLM) and Se-methyl selenocysteine (MSC) [28,43,44,45,46]. These effects were associated with enhanced antitumor activity of anticancer drugs in multiple tumor types [43,44,45]. Based on the data generated, it was postulated that the downregulation of the oncogenic miRs and HIFs in tumor cells and the associated microenvironment using selenium will result in the multidrug sensitization of tumor cells. Therapeutic application of the optimally nontoxic doses and schedule of SLM or MSC as a pleiotropic master regulator of drug resistance biomarkers may offer the potential to avoid the need to evaluate multiple agents with independent targets and overlapping toxicity profiles. The proof of concept is being validated in patients with advanced ccRCC using expanded doses of SLM in sequential combination with axitinib [47,48]. Validation of the therapeutic role of SLM in ccRCC would provide the rationale for the future development of additional combinations in ccRCC and other cancers with similar SLM target expressions.

## 2. Results

### 2.1. miR Expression Profile of ccRCC Cell Lines Relative to Normal Kidney Cells

The relative expression levels of miR-17, -19a, -34a, -155, and -210 in five ccRCC cell lines without sarcomatoid cell differentiation—OS-RC2, CAKI-1, 786-0, RCC4, and RCC4/VHL—and five cell lines with sarcomatoid cell differentiation—RCJ41T1, RCJ41T2, RCJ41M, UOK127, and UOK276—were quantified relative to RPTEC, the designated normal epithelial cell line, and the results are displayed in Figure 1. In addition, the relative miR expressions of each cell line were normalized using its respective U6 levels. In all ten cell lines, miR-17 and miR-19a, which are tumor suppressor genes, were not significantly upregulated above levels expressed in normal kidney cells. In contrast, miR-34a and miR-155 were significantly upregulated in eight out of ten cell lines, irrespective of the HIFs isoform expressions, *VHL* functional status, or sarcomatoid differentiation. In the RC2 cells that expressed HIF1α and mutant *VHL*, none of the five miRs were significantly upregulated. In the sporadic chromophobe UOK276 cells with a *TP53* missense mutation, only miR-34a was significantly upregulated.

### 2.2. miR Profile in ccRCC Biopsies and Matching Normal Kidney Tissue

The expression levels of miR-17, -19a, -34a, -155, and -210 were assessed in kidney tumors with matching normal kidney biopsies obtained from six patients (four ccRCC, one unclassified#1411, one Papilary#1418 with the two ribosomal RNAs present for both kidney tumor (STP) and normal kidney tissue (STN) samples; Figure 2, Appendix A). The data indicated that the expression of miR-17 and -19a, which were presumed tumor suppressor miRs, were expressed at lower levels than miR-34a, -155, and -210. The average expression intensity of miR-210, miR-155, and miR-34a in the six tumor samples were 4.7, 4.4, and 2.7, respectively. Since only one tumor from each of the six patients was analyzed several times with reproducible results, it was likely that miRs may be regulated by different mechanisms among different tumors with the same histology. To confirm statistical significance in the expression among the five miRs in individual patients, a bigger sample size would be required. 

### 2.3. Modulation of miRs Using Biologically Targeted Agents

To assess the potential effects of treatment with biologically targeted molecules on the expression levels of TGF-β, PD-L1, and VEGF assessed using Western blotting and the five miRs, kidney tumor tissues were obtained from three ccRCC patients treated with either cabozantinib (#1436), pembrolizumab with axitinib (#1584) or ipilimumab with nivolumab (#1318). Patients were treated uniformly with SLM (5000 μg, twice daily for 14 days) and with 5 mg axitinib administered twice daily starting at day 15. The data in the three treated patients were compared to representatives of three untreated RCC tumors, i.e., two ccRCC and one papillary, shown in #1418 in Figure 3. Patients treated with cabozantinib, which targets VEGFR, achieved a partial response, demonstrating a trend of lower expression of VEGF, as well as miR-17, -19a, -34a, -155, and -210. The patient treated with pembrolizumab and axitinib, which target programmed death-1 (PD-1) and VEGFR, respectively, achieved a partial response, and the levels of TGF-β, VEGF, and the five miRs were significantly downregulated. Similarly, a papillary patient treated with ipilimumab and nivolumab, which target cytotoxic T-lymphocyte associated protein 4 (CTLA4) and PD-1, respectively, achieved stable disease and the levels of all biomarkers were downregulated. Since there were no tumor tissues available prior to therapy, the observed effects may not have been directly treatment-related.

## 3. Discussion

Although significant advances were achieved in the treatment of patients with advanced renal cell carcinoma, the lack of cures and resistance to antiangiogenic and immunotherapies-based combinations continue to represent major clinical challenges. Based on clinical results to date, there is an unmet need to identify additional druggable targets. Based on data generated in two ccRCC cell lines, namely, RC2 and 786-0, demonstrating that miR-210 and miR-155 are critical therapeutic targets and can be downregulated by a defined dose and schedule of selenium [44], this study was carried out to generate additional data that was aimed toward further defining the expression levels and intensities of miRs altered in ccRCC tumor cell lines and primary kidney tumor biopsies with and without sarcomatoid differentiation. Although multiple types of altered miRs were identified, the focus of our investigation was on miR-17, -19a, -34a, -155, and -210. 

The oncogenic miR-155 and -210 are ubiquitously overexpressed oncogenes in a variety of advanced cancers and function as master regulators of multiple targets associated with increased tumor angiogenesis, drug resistance, and immune evasion [14,16,21,22,23,24,25,26,49,50,51,52,53,54]. Although the oncogenic miR-155 and miR-210 are considered hypoxic biomarkers, they are also overexpressed in tumor cells under normoxic conditions. In addition, miR-155 and -210 and HIFs regulate the expression of the transforming growth factor-beta (TGF-β). TGF-β is a multifunctional cytokine that exerts dual effects, functioning both as a tumor suppressor gene in normal tissues and early cancers and as an oncogene in late-stage cancer. In tumor tissues, TGF-β is reportedly regulated, in part, by HIF1α, HIF2α, miRs-155, and miR-210 [4,5,6,7,8,11,12,13,20,33,55]. The overexpression of TGF-β in tumor cells exerts dual effects: regulating the expression of PD-L1 and VEGF and affecting biomarkers associated with the modulation of immune response and evasion, including Foxp3/Treg, MDSC, Th1/Th2 balance, and macrophage cells [9,10,17,18,19,49,56,57,58]. HIFs and TGF-β are biomarkers associated with the expression of the *AXL* gene, which is highly overexpressed in tumors with a loss of *VHL* function and is associated with a poor response to biologically targeted molecules [59,60,61,62,63]. Based on published data generated by others and data generated in our laboratory, Figure 4 is presented to illustrate the potential role of miR-210, miR-155, TGF-β, and HIFs individually or collectively in the regulation of multiple targets. Data generated in our laboratory in vitro demonstrated that the downregulation of miR-210, miR-155, TGF-β, and HIFs using selenium were associated with the downregulation of PD-L1, VEGF, and several other tumor suppressor miRs, Foxp3/Treg, and *AXL* ([43,44], and unpublished data). Figure 4 also illustrates the potential role of TGF-β in inducing the epithelial-to-mesenchymal transition [64,65], P-glycoprotein [66], nuclear factor-erythroid factor 2-related factor 2, macrophage [46], glycogen synthase kinase [67], and mitochondrial lipid metabolizing enzymes that regulate the stable accumulation of lipid droplets, which is a unique histological feature associated with ccRCC [2,3]. The fact that TGF-β, miRs, and HIFs regulate multiple targets associated with increased tumor angiogenesis, immune evasion, and drug resistance highlights the unmet need to delineate the underlying mechanisms and to develop potent and selective inhibitors.

In both ccRCC cell lines with and without sarcomatoid differentiation and in the six tumor biopsies, miR-155, miR-210, and TGF-β were significantly upregulated relative to normal adjacent tissue. The data generated in the nine ccRCC cell lines and in the six biopsies showed that miR-17 and miR-19a were downregulated, while miR-34a was upregulated in cell lines with and without sarcomatoid differentiation and downregulated in the six ccRCC tumor biopsies. miR-17, -19a, and -34a were identified to function either as tumor suppressors or as oncogenes [49,53,63,68,69,70]. The function of these genes is likely determined by tumor types, cancer stage, and the relative expression levels of these miRs. The data generated in the ccRCC cell lines and tumor biopsies were consistent with previously reported results that demonstrated an inverse relationship between the TGF-β expression level and the expression level of miR-34a [14,15,16,49,50,51,52,53,56,71]. Many of the cell lines outlined in Figure 1 are being used widely in preclinical research. The quantitative and qualitative differences and similarities in the microRNA profiles between cell lines and clinical kidney biopsies as indicated in Figure 1 and Figure 2 might help direct future translational research.

The rationale for the analysis of the miR biomarkers was based on data generated in preclinical ccRCC, demonstrating that specific types of oncogenic and tumor suppressor miRs are druggable targets using a defined dose and schedule of SLM and MSC administration [44]. Although mechanisms of the modulation of the indicated biomarkers using selenium are not fully understood, the demonstrated therapeutic synergy of selenium in sequential combination with multiple anticancer therapeutics, including axitinib and sunitinib, in several preclinical in vivo models provided the rationale to evaluate the therapeutic potential of SLM in sequential combination with standard treatment in patients with advanced ccRCC and other cancers with similar selenium target expressions. The potential role of SLM in the modulation of the antitumor activity of axitinib is being evaluated in patients with advanced renal cell carcinoma with promising and encouraging results [47,48].

Normalizing the functional activity of miR-210 and miR-155 could result in the sensitization of tumor cells to a variety of cytotoxic drugs and immunotherapy used alone and in sequential combination. Targeting a single biomarker altered in heterogeneous tumor cells and their associated microenvironment is necessary but may not be sufficient to induce durable responses and cures. Standard treatment of patients with advanced ccRCC demonstrated responses in patients with and without PD-L1 expression. The evasion of cancer to the effects of these drugs is likely achieved by a close interaction between molecular and immunological biomarkers altered in tumor cells and the associated microenvironment. Although we demonstrated that the downregulation of the HIFs, miR-210, and miR-155 using a defined dose and schedule of selenium enhances the antitumor activity of multiple chemotherapeutic drugs and biologically targeted molecules in several xenograft tumors [43,44,45], the mechanisms that are predictive of the observed therapeutic benefit have not yet been fully delineated.

In summary, the results generated in ccRCC cell lines with the differential expression of HIFs isoforms, with and without sarcomatoid differentiation, and in renal cell carcinoma tumor biopsies demonstrated that miR-17 and -19a are downregulated and miR-34a, -210, and -155 are upregulated. Since HIFs and TGF-β are also upregulated in the same cell lines and tumor biopsies used for miRs analysis (unpublished data), the hypothesis is that miRs, HIFs, and TGF-β individually or collectively regulate targets implicated in increased tumor angiogenesis and drug resistance. The demonstration that HIFs, miRs, and TGF-β are druggable targets when using a defined dose and schedule of SLM provides the scientific rationale to delineate mechanisms of SLM action and the potential for the development of mechanism-based combination therapy. To this end, phase 1 clinical trial of a defined dose of SLM in sequential combination with axitinib in patients with advanced ccRCC previously treated with multiple biologically targeted agents was completed with very encouraging results [47,48]. 

## 4. Materials and Methods

### 4.1. Cell Lines

Clear cell renal cell carcinoma cell lines OS-RC2, 786-0, RCC4, RCC4/VHL, and CAKI-1 were obtained from ATCC. Sarcomatoid cell lines UOK127 and UOK276 were provided by Dr. M. Linehan., N.C.I. [34,55,72], and RCJ41T1, RCJ41T2, and RCJ41M were received from Dr. T.H. Ho, Mayo Clinic [73]. The 11 cell lines, including RPTEC as the control, were cultured in T-75 flasks; once the cells reached 70–80% confluency, they were plated onto 60 mm diameter Petri dishes. Upon reaching 70–80% confluency, each dish was removed from the 37 °C incubation. Existing cell culture media was aspirated, and the dishes were rinsed with 2 mL of 4 °C phosphate-buffered saline (PBS) (Gibco, Invitrogen, Waltham, MA, USA). The PBS was aspirated, and the cells were rinsed once more with 1 mL PBS. The cell monolayers were scraped, suspended in PBS, and then transferred to 1.5 mL microcentrifuge tubes. The tubes were centrifuged at 7000 RPM for 1 min to form pellets. The supernatant was aspirated and 700 µL of TRIzol Reagent (Ambion, Life Technologies, Carlsbad, CA, USA) was added to the cell pellets to form lysate. The process for RNA isolation and cDNA synthesis is described below.

### 4.2. RNA Isolation

A total of 140 µL of chloroform (Thermo-Fisher Scientific, Fair Lawn, NJ, USA) was subsequently added to the TRIzol lysate, and the samples were then centrifuged at 12.0 g for 10 min under 4 °C incubations. The aqueous phase resulting from the centrifugation was then transferred to a new 1.5 mL centrifuge tube with 350 µL of isopropanol (Sigma-Aldrich, St. Louis, MO, USA). The tubes were then centrifuged at 12.0 g again for 10 min under 4 °C incubations. The supernatant was discarded and 700 µL of RNase-free 70% ethanol was added to the remaining pellet and centrifuged at 16.1 g for 5 min under 4 °C incubations. The supernatant was discarded, and the tubes were left to dry inverted on Kimwipes (Kimberly-Clark Worldwide, Inc., Roswell, GA, USA) for 15 min. After drying, 40 µL of RNase free ddH_2_O was added to the RNA pellets and the RNA concentration was measured using a ThermoFisher Scientific NanoDrop 2000 C spectrophotometer (ThermoFisher Scientific, Waltham, MA, USA). The RNA samples were subsequently run in 1% agarose gel to inspect the 18 s and 28 s bands, assessing the fidelity of the extract and the extent of degradation. The procedure was carried out three times to ensure a biological triplicate for statistical analysis.

### 4.3. cDNA Synthesis

The isolated RNA specimen was converted to cDNA using the Takara miRX kit (Takara Bio, Tokyo, Japan). Each PCR tube had a total volume of 10 µL, where 2.5 µL consisted of miRQ Buffer (2X), 0.5 µL of mRQ Enzyme Mix, and the remainder was a dilution of RNase free distilled water and RNA to yield a total RNA concentration of 2 µg/µL for all samples. The thermocycler used was a ThermoFisher Scientific GeneAmp™ PCR System 9700. The thermocycle settings for cDNA conversion were set to 37 °C for 60 min and 85 °C for 5 min. After incubation, the newly synthesized cDNA samples were stored at −80 °C.

### 4.4. Real-Time Quantitative Polymerase Chain Reaction (RT-qPCR)

The instrument used for carrying out RT-qPCR was a Bio-Rad CFX Connect Real-Time PCR Detection System (BioRad Laboratories, Inc., Hercules, CA, USA). The thermal cycling profile consisted of 95 °C for 3 min, followed by 45 cycles of denaturation at 95 °C for 10 s and annealing at 60 °C for 30 s. The melt curve was analyzed to ensure the fidelity of the qPCR products. The primers utilized for experimentation were 0.1 µM forward miR primer and 0.1 µM universal primer, which was the Takara mRQ 3′ Primer. The master mix consisted of a 1:1 ratio of autoclaved ddH_2_O and Takara TB Green Premix Ex Taq II (Tli RNase H Plus). A total of 1 µL of 2 µg/µL of cDNA was added to each well. The control for the RT-qPCR analyses was 0.1 µM forward and 0.1 µM reverse primers of Takara U6. The fold expressions of each sample relative to the control group, i.e., RPTEC, were calculated using the ∆∆CT algorithm. The sequences of miRNA forward primers used in the study are provided in Table 1 below and the miRX reverse primer purchased from the Takara company was used.

## Figures and Tables

**Figure 1 ijms-23-05604-f001:**
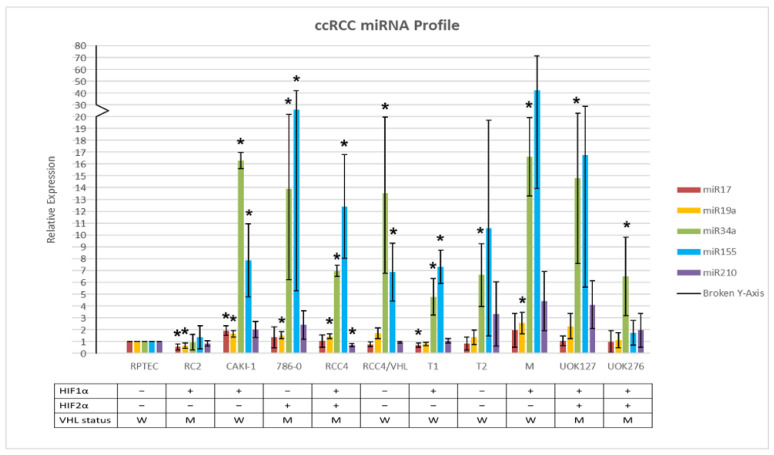
MicroRNA profile of ccRCC cell lines with and without sarcomatoid differentiation. RCJ41T1 was abbreviated as “T1”, RCJ41T2 as “T2”, and RCJ41M as “M”. Additional information was supplemented to specify whether each cell line expressed HIF1α and/or HIF2α and whether the Von Hippel–Lindau (*VHL*) factor was wild-type (W) or mutant (M) by nature. All eleven cell lines were cultured simultaneously and harvested for RNA isolation with a sample population of n = 3. An asterisk (*****) denotes a p-value of less than 0.05.

**Figure 2 ijms-23-05604-f002:**
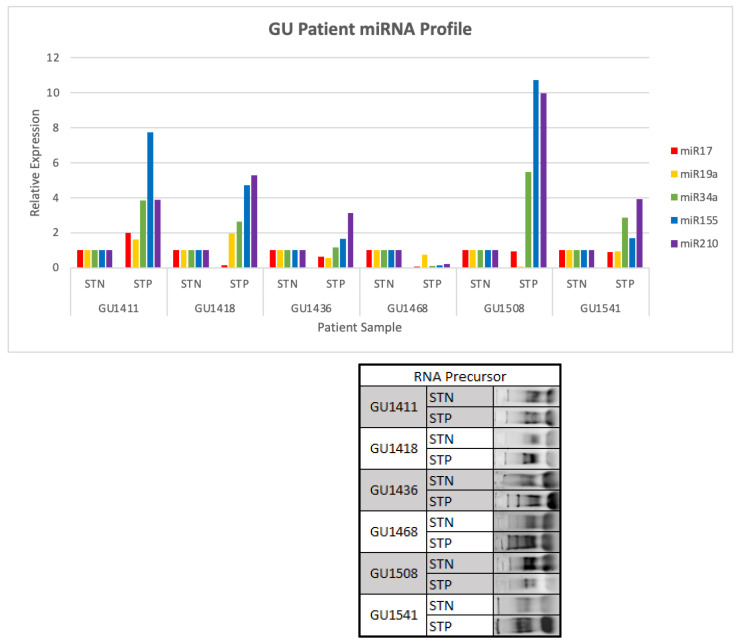
Quantitative expression of miRNA-17, -19a, -34a, =155 and -210 of six kidney tumors (STP) relative to matching adjacent normal kidney tissues (STN) from patients with advanced renal cell carcinoma and the table underneath it showed the RNA gel pictures with the two ribosomal RNAs present for both the STN and STP samples.

**Figure 3 ijms-23-05604-f003:**
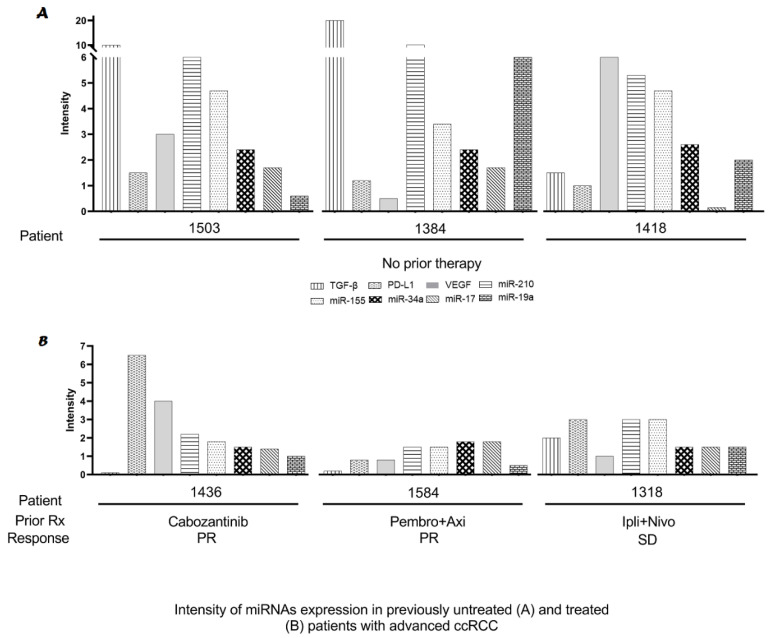
Comparative microRNA expression in 3 previously untreated (**A**), naïve and treated (**B**) ccRCC patients. The intensity was defined as the fold increased expression for each biomarker in tumors above normal controls. PR, partial response defined as a decrease in tumor size by 50% or more compared with the baseline before therapy; SD, stable disease with no change in tumor size from the baseline; Rx, the type of therapy administered. Pembrolizumab, which is an anti-PD-1 inhibitor, is abbreviated as “Pembro”; axitinib, which is a vascular endothelial growth factor receptor inhibitor, is abbreviated as “Axi”; Ipilimumab, which is an anti-cytotoxic T-lymphocyte-associated protein 4 (anti-CTLA4) inhibitor, is abbreviated as “Ipli”; and Nivolumab, which is an anti-PD-1 inhibitor, is abbreviated as “Nivo”. Untreated patients have a higher expression level of various miRNAs, TGF-β, PD-L1, and VEGF than patients treated with various medications, resulted in either partial response (PR) or stable disease (SD).

**Figure 4 ijms-23-05604-f004:**
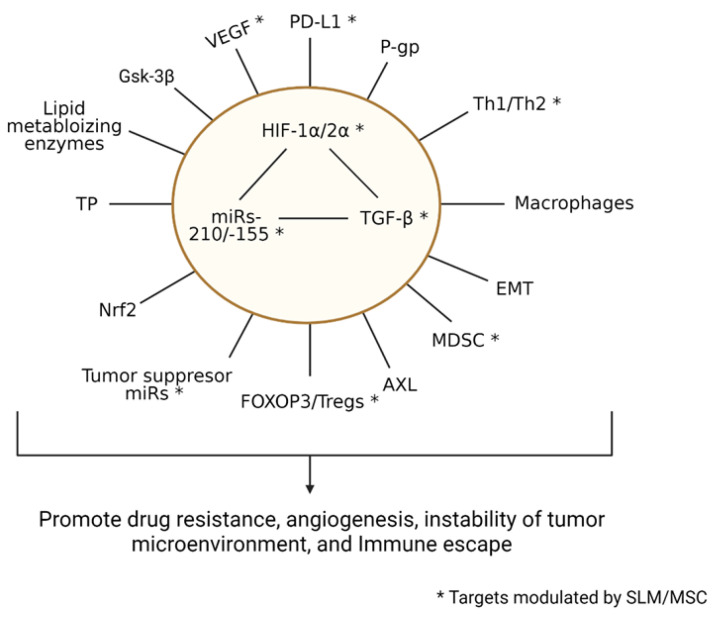
Schematic representation of the potential interaction of HIFs, miR-210, miR-155, and TGF-β that could result in cooperative regulation of reported biomarkers altered in ccRCC and implicated in an unstable tumor microenvironment, increased tumor angiogenesis, and drug resistance. * Targets modulated by either SLM or MSC.

**Table 1 ijms-23-05604-t001:** Forward and reverse primers sequences used in the RT-qPCR experiment.

miRNA	Forward Primer Sequences	Reverse Primer Sequences
Hsa miR-17 (5′)	CAAAGTGCTTACAGTGCAGGTAG	
Hsa miR-19a (3′)	TGTGCAAATCTATGCAAAACTGA	
Hsa miR-34a (5′)	TGGCAGTGTCTTAGCTGGTTGT	
Hsa miR-155 (5′)	TTAATGCTAATCGTGATAGGGGTT	
Hsa miR-210 (3′)	CTGTGCGTGTGACAGCGGCTGA	
U6	CTCGCTTCGGCAGCACAT	TTTGCGTGTCATCCTTGCG

## Data Availability

Data available on request from the corresponding author.

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
