# Peer review of "MicroRNA Expression in Clear Cell Renal Cell Carcinoma Cell Lines and Tumor Biopsies: Potential Therapeutic Targets"

_ijms, 2022, doi:10.3390/ijms23105604_

Round 1

Reviewer 1 Report

The manuscript "MicroRNAs expression in clear cell renal cell carcinoma cell lines and tumor biopsies; potential therapeutic targets", written by Swearson S, Rataan AO, Eliason S, Amendt BA, Zakharia Y, Salem AK, Ho T. and Rustum YM. presents the analysis of expression of a set of genes and miRNAs in cell lines and tumor biopsies of renal carcinoma.

Although the authors did some changes and responded to several comments, there are still some remarks.

There are still comparing miRs one with other: "Unlike the high expression of miR-210 in primar kidney tumor tissues, this miR is expressed at lower level than miR-155 in all 10 cell lines, although  miR-210 appears to be expressed at higher level in cells with sarcomatoid differentiation compared to cells with no sarcomatoid differentiation."

line 123: "The data indicates that the expression of miRs-17 and 19a, presumed tumor suppressor miRs, are expressed at lower levels than miRs-34a, -155 and-210."

If expression in sarcomatoid and nonsarcomatoid differentiation is compared, it should be statistically confirmed.

Describing cell lines, they are compared with primary tumors which are not yet described.

Figure 2 (in fact Table) is added. It misses explanation of abbreviations, explanations of the method and statistics. On the other side, part of these data could be put in supplemental material as there is no need of numbers of PCR cycles in the manuscript.

Line 140: "The data in Figure 3 indicate that TGF-, PD-L1 and VEGF protein are expressed in 79%, 50%, and 29%, respectively." This statement needs more explanation, what do these percentages mean. Densitometry could be done.

In the Discussion still 10 or 11 patient samples are discussed, although they are not any more in the results. (Also protein analysis is presented for all the samples, without the comment) In the Discussion still all previously analyzed 10 or 12 biopsies are commented, although only 6 were finally analyzed.

Sentence reconstruction line 127: Since only one tumor with a matching normal kidney tissue sample were obtained from individual patient were analyzed several times with reproducible results, it is not possible to do informative statistical analysis.

Line 252 "Normalizing the functional activity of HIFs, miRs-210/-155, and TGF-β combined with the stabilization of tumor vasculature by selenium" Explanation of the statement

line 141: sentence reconstruction "Consistent with the data presented in Figure 2 above, the incidence TGF- is over expressed at higher level in comparison with PD-L1 and VEGF"

sentence reconstruction: Although this patient achieved durable  partial response to the SLM/axitinib combination, the result generated in one patient, however, do not confirm that miRs are druggable target by SLM alone and in combination with axitinib. Since  clinical data demonstrated responses to targeted therapy was documented independent of PD-L1  and VEGF, the intended target, confirming in larger sample size that miRs are SLM/axitinib druggable targets will be therapeutically important.

Materials and methods:

Correct the sentence: "Total protein was extracted from paired normal and tumor resection specimens from 6 patients."

To my opinion, analysis of cell lines and biopsies on Mirs and gene expression contain publishable results. Data on patient therapy and responses do not have reliable controls for conclusions. Description of signaling pathways is superficial.

Author Response

We appreciate all of the reviewers' time and effort in reading our manuscript and providing important feedback and ideas. The manuscript has been revised in response to the reviewers' suggestions

Reviewer 2 Report

The section, Case report, should be omitted as the authors do not show any scientifically sound data.

Author Response

We appreciate all of the reviewers' time and effort in reading our manuscript and providing important feedback and ideas. The manuscript has been revised in response to the reviewers' suggestions.

Reviewer 3 Report

The article has undergone significant revision. Research methods have been expanded, all comments of the reviewers have been taken into account

Author Response

We appreciate all of the reviewers' time and effort in reading and revising our manuscript 

Round 2

Reviewer 1 Report

The manuscript "MicroRNAs expression in clear cell renal cell carcinoma cell lines and tumor biopsies; potential therapeutic targets", written by Swearson S, Rataan AO, Eliason S, Amendt BA, Zakharia Y, Salem AK, Ho T. and Rustum YM. presents the analysis of expression of a set of genes and miRNAs in cell lines and tumor biopsies of renal carcinoma.

Although the authors did some changes and responded to several comments, there are still some remarks.

Although now only 6 samples of biopsy tissue are described, considering RNA expression, there are 12 or 14 protein samples analyzed, and results and materials and methods are not consistent.

There is still the main part of the Discussion dedicated to the therapy and the effect of the selenium, and that was not the topic of the manuscript. Now there are only 3 samples of tumors (with and without therapy), and without more samples and statistics it cannot be concluded anything.

To my opinion, analysis of cell lines and biopsies on Mirs and gene expression contain publishable results. Data on patient therapy and responses do not have reliable controls for conclusions.

Author Response

We appreciate your time and effort in reading and revising our manuscript.

Reviewer 2 Report

I have no further comment to make.

Author Response

(The authors gave the same response as above.)

Round 3

Reviewer 1 Report

I have no more major comments.

Figure 2 should be corrected.

This manuscript is a resubmission of an earlier submission. The following is a list of the peer review reports and author responses from that submission.

Round 1

Reviewer 1 Report

The manuscript "MicroRNAs expression in clear cell renal cell carcinoma cell lines and tumor biopsies; potential therapeutic targets", written by Swearson S, Rataan AO, Eliason S, Amendt BA, Zakharia Y, Salem AK, Ho T. and Rustum YM. presents the analysis of expression of a set of genes and miRNAs in cell lines and tumor biopsies of renal carcinoma. Expression of HIF2 alpha, HIF 1 alpha,TGF beta, and miR-210, miR-155, miR-34a, miR-17 and miR-19a were analyzed in 10 renal tumor cell lines, 14 tumor biopsies and corresponding normal tissues, as well as in several samples of patients receiving therapy. One sample is presented as a case report, analyzed before the therapy, after 14 days and after 3 months. The patient was treated with combination of axitinib and a selenium compound. As the treatment downregulated miRNAs, the authors conclude that they are also druggable targets in selenium therapy of renal carcinoma.

The authors used RT qPCR to analyze the expression of several genes and miRs. There are no data on miR normalization.  To normalize miR expression, the expression of each miR could be compared with the expression of specific miR which is known to have stable expression (or with rRNA amplification). On the other side, the authors compare miR expression with each other. I think there is no need of such comparisons, as each miR can be regulated by different transcription factors and can be independently regulated. There is no statistical analysis of expression values. In presentation of patient gene expression there are no data of standard deviations. Figure 4 is not explained enough; it is not clear what does it mean PR, SD, Rx.

In Discussion expression of miR-155, miR-210 and TGF beta are compared. It could be possible to do correlation analysis to make such statements and describe potential targets of deregulated miRs. miRs, HIF and TGF beta are described as interactive, but more data on their interaction should be presented, as their regulation can be a consequence of some other processes. miRs do not have to be direct targets of the therapy. In description in Figure 6 it is said that HIF, TGF beta and miR could potentially interact resulting in cooperative regulation. These data are not supported by citations and mechanism explanation.

There is no data on ethics approval.

Minor comments:

line 18: line

line 49: including

line 108: Figure 1: explanation of W, M, experimental settings...

lines 117, 134: each miR has its own regulation (unless in regulation circle) and there is no sense of their comparison

line 126: indecently

line 143: Figure 3: biomarkers in comparison with those of normal kidney

lines 154, 180, 170: there is no unit mcg; units should be written separately from numbers

line 158: missing part of the sentence

lines 159, 162: target; explanation of partial response

line 170: data were

line 207: FOXP3, explanation of MDSC, sentence correction

line 255: to predict

Author Response

Dear Reviewer 1,On behalf of all the authors I would like to thank you for your excellent comments. The reviewer comments were addressed in totality and incorporated in the attached revise manuscript . Hope the manuscript is now acceptable for publication.With many thanks

Reviewer 2 Report

In this manuscript, the authors presents the initial results of expression profiles of miRNAs and their target mRNAs in clear cell renal cell carcinoma cell lines and tumor biopsies. Although valuable patients’ samples were used, the number of samples for each assay is very small, which precludes the conclusion to be drawn. More specific comments are listed below:

Major points:

[1] The sample is too small. The authors must include the sample size. Also, the authors should compare their results to the publicly available data of larger cohorts, such as those of Cancer Genome Atlas (TCGA).

[2] Figure 3. The definition of intensity of expression must be described in details.

[3] Figure 4. The authors must provide the protein expression of the target genes (i.e., Western blotting, ELISA).

[4] Figure 5. This is n = 1. The authors must demonstrate the findings with much much more patients’ samples.

Minor points:

(1) The sample size for each assay result is missing in the figure legends.

(2) The figure legends must include more information regarding the data shown.

(3) The details about statistical analysis performed are missing.

Author Response

Dear Reviewer 2,On behalf of all the authors I would like to thank you for your excellent comments. Your comments were addressed in totality and incorporated in the attached revise manuscript . Hope the manuscript is now acceptable for publication.With many thanks

Reviewer 3 Report

The article is devoted to an actual topic. new approaches in molecular mechanisms in the development of renal cell carcinoma are extremely relevant. The mechanisms of epigenetic regulation allow not only to expand the understanding of the pathogenesis of cancer, but also to form the basis for the appointment and testing of new targeted drugs

Author Response

Dear Reviewer 3,

Thank you very much for your comments and feedback.

Round 2

Reviewer 1 Report

The manuscript "MicroRNAs expression in clear cell renal cell carcinoma cell lines and tumor biopsies; potential therapeutic targets", written by Swearson S, Rataan AO, Eliason S, Amendt BA, Zakharia Y, Salem AK, Ho T. and Rustum YM. presents the analysis of expression of a set of genes and miRNAs in cell lines and tumor biopsies of renal carcinoma. The authors improved the manuscript, although I still have some comments.

The authors used RT qPCR to analyze the expression of several genes and miRs. Although the kit used for miRNA had U6 as an internal standard, there is no data on primers and genes used for standards for mRNA genes.  The authors added a table with comparison of miRNA expression among normal and tumor cell lines, but still compare miR expression with each other. I think there is no need of such comparisons, as each miR can be regulated by different transcription factors and can be independently regulated. There is no data on the number of replicates. Figure 4 is better described but still needs explanation what is Ipli, Nivo, Pembro, Axi etc.

In Discussion expression of miR-155, miR-210 and TGF beta are compared. As a conclusion, it is stated that selenium compounds could influence miRs expression. It is possible that these data are proved in other publications, but this case report cannot be taken as a proof:  beside the fact that it is just one case, miRs were decreased after treatment with combination of selenium compound and axitinib and there are no controls with monotherapy.  

There is still no data on ethics approval.

Minor comments:

µg instead of mcg

units should be written separately from numbers

line 197: correction

line 242: epithelial mesenchymal transition and macrophage are not molecules

Author Response

(The authors gave the same response as above.)

Reviewer 2 Report

The authors failed to address this reviewer's previous comments. Also, the point-by-point responses are not provided.

Author Response

(The authors gave the same response as above.)

Round 3

Reviewer 1 Report

Although the authors corrected the manuscript and responded to comments, there are still several issues to be discussed.

The authors still often compare expression of different miRs with one another – I think that each miR has unique expression and biology (lines 32, 123, 139 etc.) and should be compared with normal controls. Describing tumor biopsies, it could be understood that only 6 samples have good RNA quality. I think that in that case only these samples should be taken into account.

Additionally, Western blot analysis of paired patient samples is presented. It is not described in the Results, neither in Materials and methods, there are no data and discussion of these experiments. Fig. 4 presents expression of miRNA and several genes in samples from untreated and treated patients. It should be an example of the effect of the selenium treatment, but it is not presented in the figure.

Two paragraphs in the Discussion are dedicated to description and effects of selenium compounds on renal carcinoma. That is not the topic of the manuscript. Selenium compounds were used in combination with other therapies and in small number of cases and although they could improve therapy, presented data are not proofs and do not lead to the presented conclusions. Possibly, more details on the relations between miRs and their targets and proteins involved (although it is said that they regulate TGF beta), signaling pathways and mechanisms of tumorigenesis in which they are involved could be described.

Minor comments:

5000 µg = 5 mg

line 128, 129: sentence reconstruction

line 141, 198: sentence reconstruction

line 215: missing reference

Reviewer 2 Report

[1] The authors' response to Comment #1: "It is acknowledged that the sample size in some cases is small, in those cases it is indicated that for more definitive conclusions, results should be confirmed in larger sample size."

This is a scientific manuscript, especially dealing with molecular biology in relation to clinical study. There is no scientific evidence to trust the results of n = 1 sample size.

[2] Western blotting. Why did the authors run all the samples on the same SDS-PAGE gel to demonstrate the changes in protein expressions of all samples? The images shown are heavily cropped and adjusted. It is now a common practice to provide the whole membrane image as supplementary data. Also, the Material and Methods section for Western blotting assay is missing. The catalog and batch number of each antibody used must be provided along with the amount of antibody used for each assay.